# Antimicrobial susceptibility of western Canadian *Brachyspira* isolates: Development and standardization of an agar dilution susceptibility test method

D. G. R. S. Kulathunga[1], John C. S. Harding[2], Joseph E. Rubin [ID][1]*

1 Department of Veterinary Microbiology, University of Saskatchewan, Saskatoon, Saskatchewan, Canada,
2 Department of Large Animal Clinical Sciences, University of Saskatchewan, Saskatoon, Saskatchewan, Canada

* joe.rubin@usask.ca

**Data Availability Statement:** This manuscript has been written and compiled in such a way that all data is presented within. The data used for constructing our standard curve is included as

## Abstract

The re-emergence of *Brachyspira*-associated disease in pigs since the late 2000s has illuminated some of the diagnostic challenges associated with this genus; notably, the lack of standardized antimicrobial susceptibility testing (AST) methods and interpretive criteria. Consequently, laboratories have relied heavily on highly variable in-house developed methods. There are currently no published investigations describing the antimicrobial susceptibility of *Brachyspira* isolates collected from pigs in Canada. The first objective of this study was therefore to develop a standardized protocol for conducting agar dilution susceptibility testing of *Brachyspira* spp., including determining the optimal standardized inoculum density, a key test variable that impacts test performance. The second objective was to determine the susceptibility of a collection of western Canadian *Brachyspira* isolates using the standardized methodology. After assessing multiple media, an agar dilution test was standardized in terms of starting inoculum ($1–2 \times 10^8$ CFU/ml), incubation temperature and time, and assessed for repeatability. The antimicrobial susceptibility of a collection of clinical porcine *Brachyspira* isolates (n = 87) collected between 2009–2016 was then determined. This method was highly reproducible; repeat susceptibility testing yielded identical results 92% of the time. Although most of the isolates had very low MICs to the commonly used antimicrobials to treat *Brachyspira*-associated infections, several isolates with elevated MICs (>32 µg/ml) for tiamulin, valnemulin, tylosin, tylvalosin, and lincomycin were identified. Overall, this study underscores the importance of establishing CLSI approved clinical breakpoints for *Brachyspira* to facilitate the interpretation of test results and support the evidence-based selection of antimicrobials in swine industry.

## Introduction

*Brachyspira* is a genus of Gram-negative, aerotolerant-anaerobic spirochetes that grow at high incubation temperatures (39–42°C). These organisms are difficult to work with because they

S1 Table, Table 2 includes all data used to determine the effect of concentration on MIC, Table 5 contains all MIC data and finally all sequence data is now uploaded to GenBank and the accession numbers (BankIt2688837 accession numbers OQ728818-OQ728904) are listed within the text.

**Funding:** JER - Swine Innovation Porc project #1344 The funders had no role in study design, data collection and analysis, decision to publish, or preparation of the manuscript.

**Competing interests:** The authors have declared that no competing interests exist.

often do not produce surface growth on agar plates, and when they do discrete colonies are typically not present. On solid media, growth is recognized as hemolytic zones on blood agar plates. Swine dysentery, which is characterized by muco-hemorrhagic diarrhea, was first described in 1921 in the United States, although it was not until the early 1970s that *B. hyodysenteriae* was isolated [1, 2]. Recently, this disease has also been identified in association with the emerging species *B. hampsonii* and *B. suanatina* [3]. Swine dysentery affects grow-finisher pigs and is one of the most economically damaging diseases associated with *Brachyspira* species [4]. In contrast, *B. pilosicoli* has a broader host spectrum including pigs, other domestic animals, wildlife, and humans and can cause porcine intestinal spirochetosis in pigs within 7 to 14 days post-weaning [5, 6].

Anecdotally, the occruence of swine dysentery has re-emerged in the late 2000s following a period of quiescence in since the mid-1990's; in 2009 *B. hampsonii* was detected for the first time in Western Canada [7]. The return of swine dysentery was concurrently identified in the mid-western United States; reports from Iowa and Minnesota also identified *B. hampsonii* which has been phylogenetically divided into three clusters (I, II and III) [8–10].

The re-emergence of *Brachyspira*-associated disease has illuminated the diagnostic challenges of this genus, particularly for antimicrobial susceptibility testing which has not been standardized for the conditions under which *Brachyspira* spp. grow [11]. The Clinical and Laboratory Standards Institute (CLSI) prescribes a standardized set of test conditions for both aerobic and anaerobic bacteria including: media composition (pH, ionic composition and presence or absence of blood), incubation time, temperature, atmosphere, starting inoculum size and test endpoints [12–14]. The effects of modifying these parameters on the results of susceptibility tests are well recognized and were perhaps best described in a 1971 report which formed the basis for an international effort to develop test standards [15]. Particularly germane for *Brachyspira* are the effects of inoculum size on MIC; because these organisms don't form colonies and enumeration of live cells/ml (CFU/ml) is difficult, determining culture density is challenging. Furthermore, *Brachyspira* spp. are recognized to grow irregularly in liquid media [11]. When conducting a broth micro-dilution test it can be difficult to differentiate whether a culture didn't grow because it was inhibited by the antimicrobial or because of insufficient organism density to grow.

Determining the antimicrobial susceptibility of *Brachyspira* relies heavily on in-house developed methods each of which may yield different results, making comparisons between labs impossible. This was exemplified by a 2005 study where eight European laboratories participated in a ring test of *Brachyspira* diagnostics and the results between labs were inconsistent as a standardized methodology was not used [11, 16]. A follow up ring trial reported overall agreement of 90% between labs, and depending on the drug, 79%-97% of MICs determined were within the pre-determined ranges [17]. This study reported differences in assay repeatability between species, ≥ 80% of MICs within the expected range for *B. pilosicoli*, however, for *B. hyodysenteriae* inter-laboratory results were less consistent [17]. While this investigation represents an advancement in the field, the authors did not provide a method for quantitatively measuring the starting inoculum. This follow-up study also included the strain B78[T] as a quality control, allowing the observed MICs for this strain to be compared to previously observed reference values [17]. However, as this investigation did not include *B. hampsonii*, further study is required to validate this method for this newly emerged pathogen [17]. Standardized antimicrobial susceptibility test guidelines are published by EUCAST and the CLSI; these prescribe parameters including test media including cation concentration and pH, incubation temperature, time and atmosphere, size of bacterial inoculum tested and in the case of disc diffusion, the antimicrobial content of each disc [13, 14, 18]. The lack of standardized interpretive criteria is another important limitation; although interpretive criteria for *Brachyspira* spp.

have been suggested, no standard resistance breakpoints have been approved by either the Clinical Laboratory Standard Institute (CLSI) or European Committee on Antimicrobial Testing [12, 18–20]. The lack of test methods and interpretive criteria are critical barriers to the evidence-based use of antimicrobials [11].

In pigs, *Brachyspira*-associated disease is treated with pleuromutilins, macrolides and lincosamides [21]. Although inter-laboratory comparisons of *Brachyspira* MICs are speculative, there is compelling within lab evidence suggesting that resistance is emerging to these drugs in pathogenic species of *Brachyspira* [22–24]. The antimicrobial susceptibility of Canadian *Brachyspira* isolates including novel species *B. hampsonii* has not been described. Therefore, the purpose of this study was to develop and standardize an agar dilution test for determining the antimicrobial MICs of *Brachyspira* spp. and to describe the antimicrobial susceptibility of an archived collection of western Canadian isolates.

## Materials and methods

### Standardized antimicrobial susceptibility test development

**Development of a standard curve relating organism concentration to an optical density.** An equation to relate a *Brachyspira* culture density to the measured optical density (OD$_{600nm}$) was developed using 3 type strains representing the species of greatest clinical importance: *B. pilosicoli* (ATCC 51139), *B. hyodysenteriae* (JXNI00000000) and *B. hampsonii* genomovar II (IDAC No 161111–01, ALNZ00000000) [25, 26]. Briefly, frozen isolates were sub-cultured on BJ agar, a commonly used selective media for working with *Brachyspira* which contains spectinomycin, spiramycin, colistin, vancomycin and rifampin [27]. Agar cultures were then transferred into brain heart infusion broth supplemented with 10% fetal calf serum (BHIS). Broth cultures were incubated for 24–48 hours at 39°C in an anaerobic jar (Anerogen TM 2.5 L, Thermo Scientific Oxide Sachet) on a magnetic stirrer to obtain fresh bacterial cultures. A series of dilutions (1:1.1, 1:1.2, 1:1.3 and 1:2–1:512) were prepared from fresh cultures to define an OD curve over a wide range of concentrations. In parallel, serial 1:10 dilutions were prepared from each initial dilution. One hundred μl of each broth culture were taken from all of the final dilutions and plated on blood agar then incubated for 42 hours at 42°C and inspected for hemolytic zones. Microsoft Excel was used to generate scatter plots relating OD$_{600nm}$ to CFU/ml, and to determine an equation describing this relationship. The relationship between OD$_{600nm}$, CFU/ml and genome equivalents/ml as measured by qRT PCR was determined and found to be consistent (S1 Table).

**Determination of the minimum inoculum required to start a culture.** *Brachyspira pilosicoli*, *B. hyodysenteriae*, and *B. hampsonii* were grown anaerobically at 42°C for 48 hrs on BJ agar [27]. Isolates were then sub-cultured into BHIS and incubated at 39°C in an anaerobic jar as described above for 24–48 hrs (usually *B. pilosicoli* and *B. hyodysenteriae* isolates grew within 24 hrs whereas *B. hampsonii* required 48 hrs of incubation). Following incubation, a drop of each culture was examined under a phase-contrast microscope at 400 magnification to confirm the presence of live, motile spirochetes. The optical density (at 600 nm) of cultures was then measured to determine the bacterial concentration (CFU/ml).

A 1:10 dilution series ($10^{-1}$ to $10^{-9}$) was made of each culture and 2 μl was inoculated onto agar. Each dilution was also sub-cultured, in triplicate, into fresh broth (1 ml into 9 ml fresh BHIS) (Fig 1). Cultures were incubated anaerobically at 39°C for 24 hrs (broths) or at 42°C for 48 hrs (agar). Following incubation, the media were inspected for growth. In the case of broth, visible turbidity compared to an uninoculated control was considered positive (growth), all turbid broths were examined microscopically to confirm the presence of motile spirochetes. In the case of solid media the presence of hemolytic zone was considered positive (growth) on

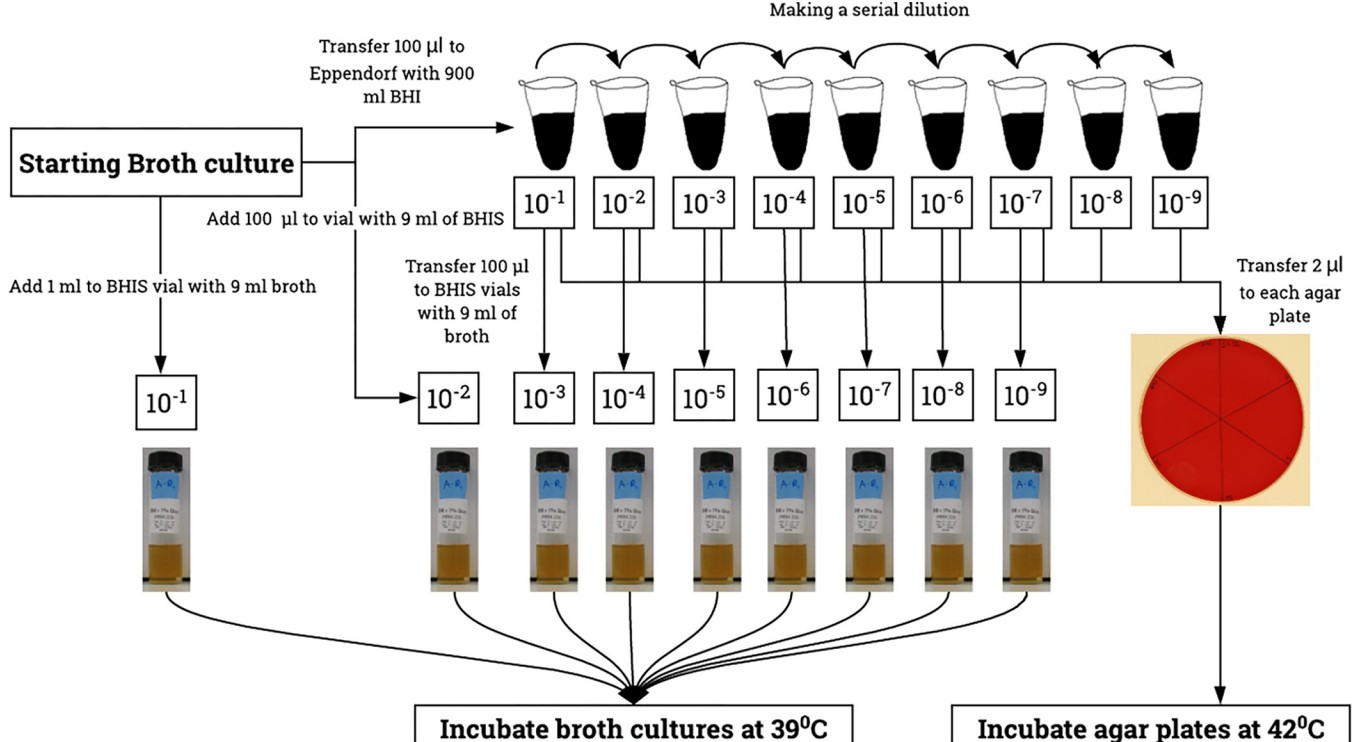

**Fig 1. Preparation of different starting inoculum sizes.** To determine the effect of starting inoculum size on the growth of different *Brachyspira* spp. in both agar and liquid media, serial dilutions ($10^{-1}$ to $10^{-9}$) of starting inoculum sizes were prepared. Then broth cultures with different inoculum sizes were inoculated by adding 100 μl of starting inoculum to brain heart infusion (BHIS) vial and by adding 2 μl to the agar media.

agar. The concentration of organisms in the most diluted starting inoculum which resulted in visible growth was then calculated using the equation derived to determine the minimum inoculum required to obtain a positive culture.

**Determining the effect of inoculum size on antimicrobial MIC.** Trypticase soy agar (TSA) + 5% sheep's blood containing antimicrobials representing four drug classes (pleuro-mutilin-tiamulin, macrolide-tylosin, β-lactam-ampicillin and quinolone-nalidixic acid) were prepared. For each antimicrobial, plates containing a series of dilutions from 0.25–128 μg/ml were made and MICs were determined for *B. pilosicoli*, *B. hyodysenteriae* and *B. hampsonii*. The organism concentration of each BHIS broth culture was determined by measuring (Thermo Scientific ND-2000 UV-Vis Spectrophotometer) the $OD_{600nm}$. A 1:5 dilution series (undiluted broth culture, 1:5, 1:25, 1:625, 1:3125) was made of each broth culture, and 2 μl was spotted onto the antimicrobial-containing plates in triplicate. Plates were incubated anaerobically for 2 days at 42°C and the lowest concentrations at which no hemolysis was observed were recorded.

By comparing our results (how MIC changes in different species with different inoculum concentrations and the repeatability of observations between replicates for each inoculum density) and conventions of the discipline (CLSI standards), an optimal starting inoculum concentration was selected.

### Antimicrobial susceptibility testing

**Isolate selection and species identification.** A total of 93 samples including 87 clinical porcine isolates collected between 2009 and 2016, four ATCC strains (*B. pilosicoli* ATCC

51139, *B. innocens* ATCC 29796, *B. murdochii* ATCC 51284, *B. hyodysenteriae* ATCC 27164) and two in-house reference strains (*B. hampsonii* 30446 and *B. hampsonii* 30599) were included in the study.

Porcine clinical isolates originated from fecal or colonic samples submitted to the *Brachyspira* diagnostic laboratory at the Western College of Veterinary Medicine, University of Saskatchewan. Briefly, upon receiving the samples, they were plated on BJ agar [27] and anaerobically incubated as described above. When hemolysis was observed, an approximately 1 cm$^2$ of agar was scraped from an isolated zone of hemolysis using a sterile bacteriological loop, macerating and then streaking it out on a sterile BJ plate. These plates were then similarly incubated at 42˚C for 48 hours and visually inspected. For each sample where *Brachyspira* grew, a total of 2–3 sub-cultures were performed to ensure that a pure culture was obtained. Thereafter, an approximately 2 cm cube of agar was sliced from an isolated β-hemolytic zone and transferred into a vial containing 10 ml of BHIS + 1% glucose (10% v:v) and incubated anaerobically at 39˚C for 24 hrs with stirring. Broth cultures were then pelleted by centrifugation, the supernatant was removed, and the pellet resuspended in 1ml of BHI + 10% glycerol for storage at -80˚C.

The clinical isolates originated from 39 different swine farms belonging to eight epidemiologically distinct production systems located in the provinces of Saskatchewan, Alberta, and Manitoba. A production system was defined as an umbrella company comprised of one or more swine farms which are independent of other companies [24]. The pigs' genotype, environmental conditions, and other natural resources (feed, manipulative materials) used in one production system may be different from other production systems [28]. The farms within a production system follow similar management practices and have pigs from a common source. The number of isolates from each production system are listed in Table 4.

The species of each isolate was identified based on partial NADH oxidase (*nox*) phylogeny as previously described [29, 30]. Briefly, DNA was purified from cell pellets of isolates grown in BHIS using a DNeasy Blood and Tissue Kit (QIAGEN) and *nox* sequences were amplified with genus-specific primers. PCR amplicons were purified using a commercial kit (BS664-250 REP, EZ-10 Spin Column PCR purification Kit, Bio Basic Canada Inc., Ontario, Canada) and were sequenced using the amplification primers. Sequences were assembled and edited using the pregap4 and gap4, and sequence alignments were performed in CLC Sequence Viewer (Version 7.7) using the ClustalW algorithm. A maximum likelihood phylogenetic tree was constructed from the aligned sequences using maximum likelihood method in MEGA (version 7.0.26) using the nucleotide substitution method with 50 bootstrap replicates [31]. Isolates were categorized as: *B. hyodysenteriae*, *B. hampsonii*, *B. pilosicoli*, *B. murdochii*, *B. innocens* based on clustering with sequences from the type strains of each species within the phylogenetic tree. Those isolates which were less than 97% similar to reference strains and fell between species clusters were categorized as non-clustering in this study.

**Agar dilution.** Ten antimicrobials, tiamulin, valnemulin, tylosin, lincomycin, tylvalosin, tetracycline, chloramphenicol, nalidixic acid, ampicillin, and amoxicillin + clavulanic acid (2:1), were selected to represent both the breadth of products used to treat swine bacterial diseases and multiple mechanisms of action. For each antimicrobial, concentrations from 0.25–128 µg/ml were prepared as per the CLSI guidelines and incorporated into TSA with 5% sheep blood which were used within 3 days [13].

Prior to susceptibility testing, isolates were cultured from freezer stocks in BHIS and incubated at anaerobically at 39˚ C for 2–3 days until turbidity was observed. Concurrently, a drop of each culture was inoculated on an antibiotic free TSA agar plate and incubated at 42˚ C anaerobically for 48 hours to ensure purity of the culture. When turbidity was observed, ODs were measured, and bacterial density was adjusted to a standard $1–2 \times 10^8$ CFU/ml. Gram-

stains were prepared from each broth to confirm the presence of Gram-negative spirochetes and the absence of contaminating organisms. A 2 μl aliquot ($2–4 \times 10^5$ CFU) of each isolate was spotted onto the antibiotic-containing (n = 300 = [(10 antimicrobials) (10 concentrations) (3 replicates)]) and antimicrobial free/positive control plates (n = 3). Plates were incubated in anaerobic jars at 42°C for 48 hours, replicates were always incubated in separate jars. Hemolysis at the inoculum site was used as the indicator of growth, and a lightbox was used to aid in the visualization of hemolysis. For antimicrobial free/positive control plates growth was always observed. The MIC was defined as the lowest antimicrobial concentration where hemolysis was not observed. In cases where MICs differed between replicates, and the difference between the highest and lowest observation was no more than a single doubling dilution, the modal value was defined as the MIC. If greater than single dilution variability in MICs was observed between replicates the isolate was retested. Because the effect of the protein synthesis inhibitors on the production of hemolysin in *Brachyspira* is unknown we were interested to know whether production of hemolysins is inhibited at a lower antimicrobial concentration than is required to inhibit the growth. To test this, in each isolate inoculation sites of both MIC and MIC + one doubling dilution were sub-cultured. Both MICs and growth (Yes/No) on sub-cultured plates were recorded. Finally, thirty-five isolates were selected, using a random number generator, for subsequent re-testing to assess the reproducibility of the assay.

**Reproducibility of the standardized agar dilution method.** Thirty-five randomly selected isolates were retested to assess the reproducibility of the assay. Observations were categorized as: complete agreement (identical between repeats), one doubling dilution difference or > one doubling dilution difference. The precision of MIC determination was defined as plus or minus 1 two-fold concentration [13], therefore, complete agreement and one doubling dilution different were considered "agreement" for the calculation of reproducibility. For isolates which were inhibited by the lowest concentration tested (MIC of $\leq 0.25$ μg/ml), the test was considered to be reproducible if both observed MICs were $\leq 0.25$ μg/ml. Reproducibility of the assay was statistically evaluated between first observation of MIC and the second observation of MIC using a parameter of correlation (Kendall's tau-b) and a measurement of agreement (Kappa).

## Results

### Development of an equation relating organism concentration to an optical density

The linear relationship of bacterial concentration and absorbance is observed with OD between 0–1 [32, 33]. Linear relationships were obtained for all three species, *B. pilosicoli*, *B. hampsonii* and *B. hyodysenteriae* (Fig 2A–2C). The calculated CFUs in all three species for each observed OD measurement were averaged to draw the equation for average (Fig 2D). According to this finding, bacterial concentration can be calculated by the following equation (equation: CFU/ml = $[(6.33 \times 10^8)^* (OD)]-8.33 \times 10^6$, where OD = optical density of a broth culture measured at 600nm (Fig 3).

### Determination of minimum inoculum size to grow *Brachyspira* on solid and liquid media

The cut-off concentration was defined as a starting inoculum sufficient to achieve visible growth on both solid and liquid media (Figs 3 and 4). For solid media, growth was observed following inoculation with as few as 1–11 CFU per spot while a substantially higher concentration ($8.7 \times 10^5–5.4 \times 10^7$ CFU/ml) was required for liquid cultures (Table 1).

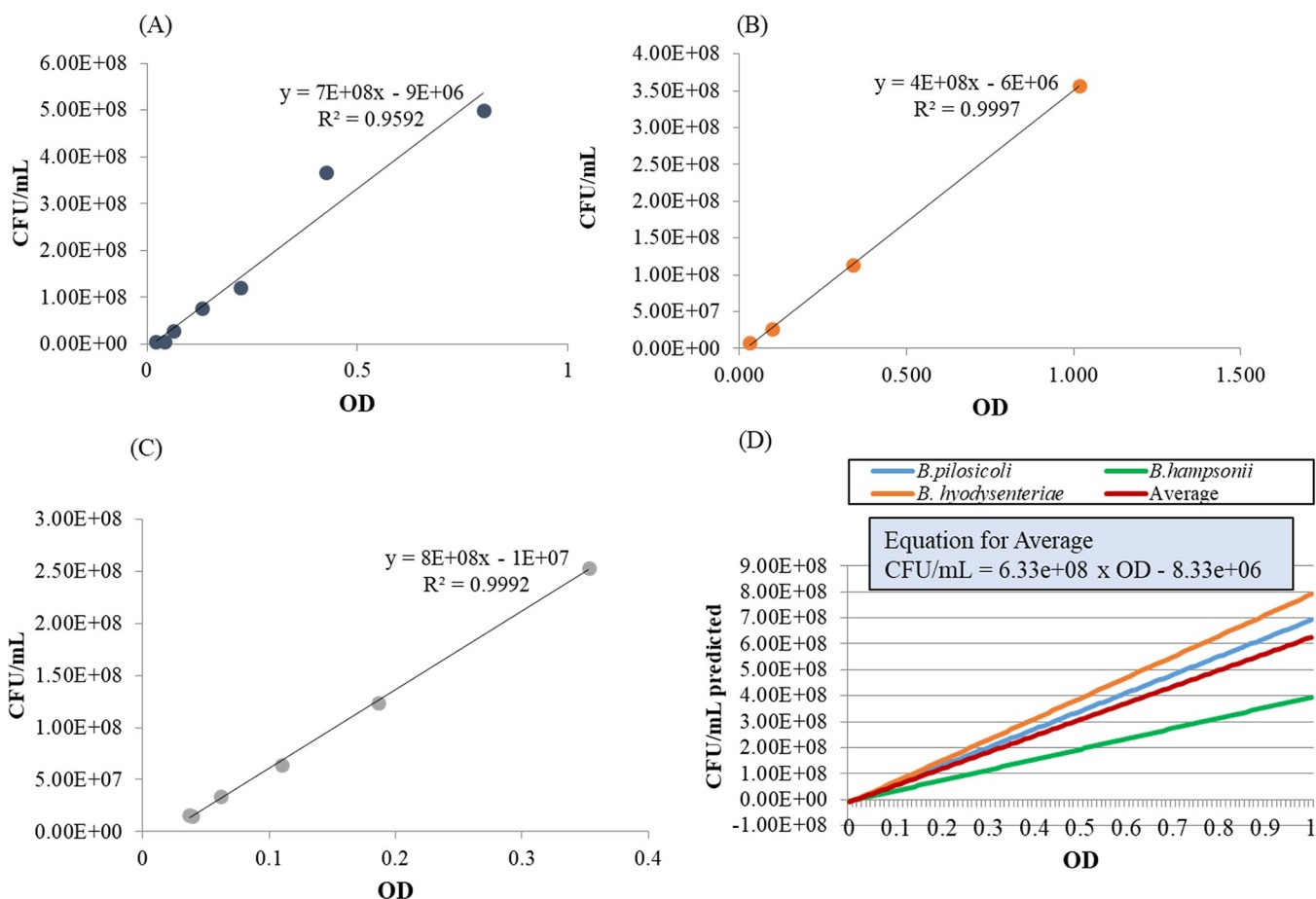

**Fig 2. The effect of optical density on determination of bacterial density (CFU/ml) in a broth culture of *Brachyspira* spp.** Relationship between optical density and colony forming unit of: (A) *B. pilosicoli* (OD), (B) *B. hampsonii*, (C) *B. hyodysenteriae*. (D) Plotted equations for all three species determined from experimental data and averaged (red line) into have an equation which can predict the bacterial density (CFU/ml) in a broth culture.

## Effect of inoculum size on observed minimum inhibitory concentration (MIC)

For all three species, consistent MICs were observed with concentrations higher than or equal to $4.4 \times 10^7$ CFU/ml for *B. hyodysenteriae*, $7.2 \times 10^7$ for *B. hampsonii* and $3.6 \times 10^7$ CFU/ml for *B. pilosicoli*. Inoculum concentrations less than this yielded inconsistent MICs between replicates (Table 2). For all species, observed MIC increased with increasing inoculum size (Table 2). Based on these results, a concentration of $1–2 \times 10^8$ CFU/ml (final inoculum of $2–4 \times 10^5$ CFU per spot on agar) was chosen to optimize assay repeatability. Furthermore, as this inoculum is the same as is prescribed by the CLSI guidelines for susceptibility testing, it will be relatively easy to integrate into current standard diagnostic procedures [13].

## Reproducibility of the standard agar dilution method

Overall, the reproducibility of this test was found to be ≥80% for all ten antimicrobials tested. The lowest reproducibility (80%) was observed for tylosin and lincomycin, while repeated valnemulin MICs were identical. Kendall's tau-b (correlation) and Kappa (agreement) were used to compare the two observations of MICs for each antimicrobial (Table 3). The lowest

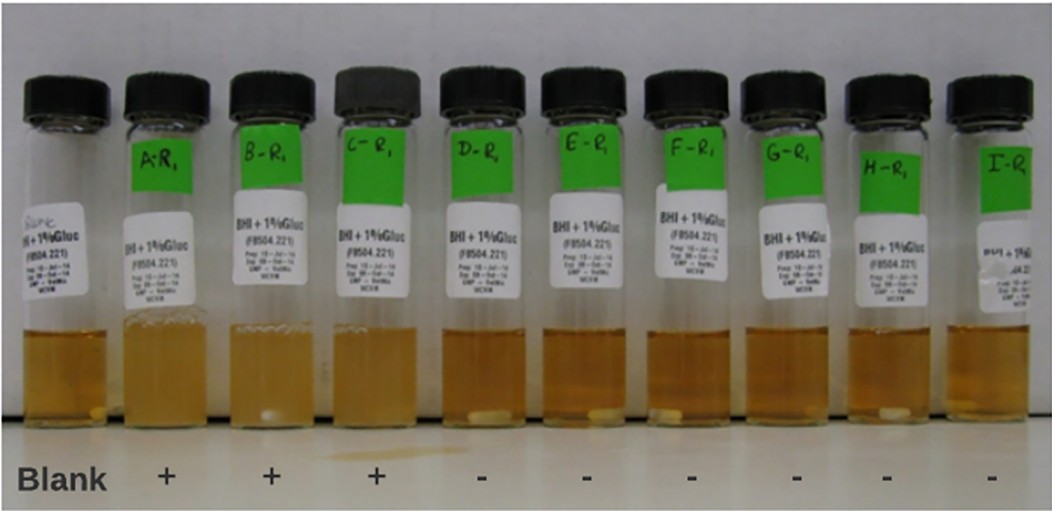

**Fig 3. The growth of *B. pilosicoli* in brain heart infusion (BHI) broth with different starting inoculum sizes.** Visible growth (turbidity = (+) growth, no turbidity = (-) growth) in brain heart infusion broth (supplemented with 1% defibrinated serum) with decreasing starting inoculum size (from left to right: uninoculated control (blank), (A) $8.7 \times 10^8$ growth, (B) $8.7 \times 10^7$ growth, (C) $8.7 \times 10^6$ growth, (D) $8.7 \times 10^5$ no-growth, (E) $8.7 \times 10^4$ no-growth, (F) $8.7 \times 10^3$ no-growth, (G) $8.7 \times 10^2$ no-growth, (H) $8.7 \times 10^1$ no-growth, (I) ~8.7 CFU/ml) no-growth.

correlations ($<0.793$) were observed in tylosin, lincomycin and nalidixic acid. Based on the Kappa values, all test agreement varied from fair (K = 0.21-.40) to substantial (K = 0.61–0.80) with the highest values observed for tiamulin, tylvalosin, chloramphenicol and ampicillin.

## Species identification

Based on *nox* phylogeny (over 765 base pairs) (BankIt2688837 accession numbers OQ728818-OQ728904) clinical isolates were identified as: *B. hampsonii* (n = 14), *B. hyodysenteriae* (n = 17), *B. pilosicoli* (n = 16), *B. murdochii* (n = 18), *B. innocens* (n = 9) and non-clustering (n = 13) (Fig 5 and Table 4). Those isolates described as non-clustering were dissimilar to type strains and could therefore not be classified into a recognized species.

## Antimicrobial susceptibility test results

A wide range of susceptibility was observed among the *Brachyspira* spp. isolates (Table 5).

Apart from *B. pilosicoli*, the majority of isolates were inhibited by low concentrations of the pleuromutilins; tiamulin inhibited the growth of 94% of the *B. hyodysenteriae* and 86% of the *B. hampsonii* isolates at the lowest drug concentration ($\leq$0.25 µg/ml). For valnemulin, all isolates of *B. hampsonii* and *B. hyodysenteriae* were inhibited at $\leq$0.25 µg/ml (Table 5). MIC$_{90}$ for tiamulin and valnemulin were 64 µg/ml and 32 µg/ml for *B. pilosicoli*, 8 µg/ml and 0.5 µg/ml for *B. murdochii* and 1 µg/ml and $\leq$0.25 µg/ml for *B. hampsonii*.

Heterogeneous MIC distributions were observed for *B. hampsonii*, *B. hyodysenteriae*, *B. murdochii*, and non-clustering group for tylosin while a bi-modal MIC distribution was observed for *B. pilosicoli*. For all spp. the MIC$_{90}$ for tylosin was $> 128$ µg/ml. A heterogeneous MIC distribution was observed in *B. hampsonii*, *B. murdochii*, *B. pilosicoli*, and *B. innocens* for tylvalosin. For tylvalosin, MIC$_{90}$ was observed to be lower in *B. hyodysenteriae* and non-clustering isolates (4 µg/ml) than for *B. pilosicoli* and *B. innocens* ($\geq$128 µg/ml). In our collection there were six isolates with very high MICs ($\geq$128 µg/ml) for tylvalosin, including a single *B.*

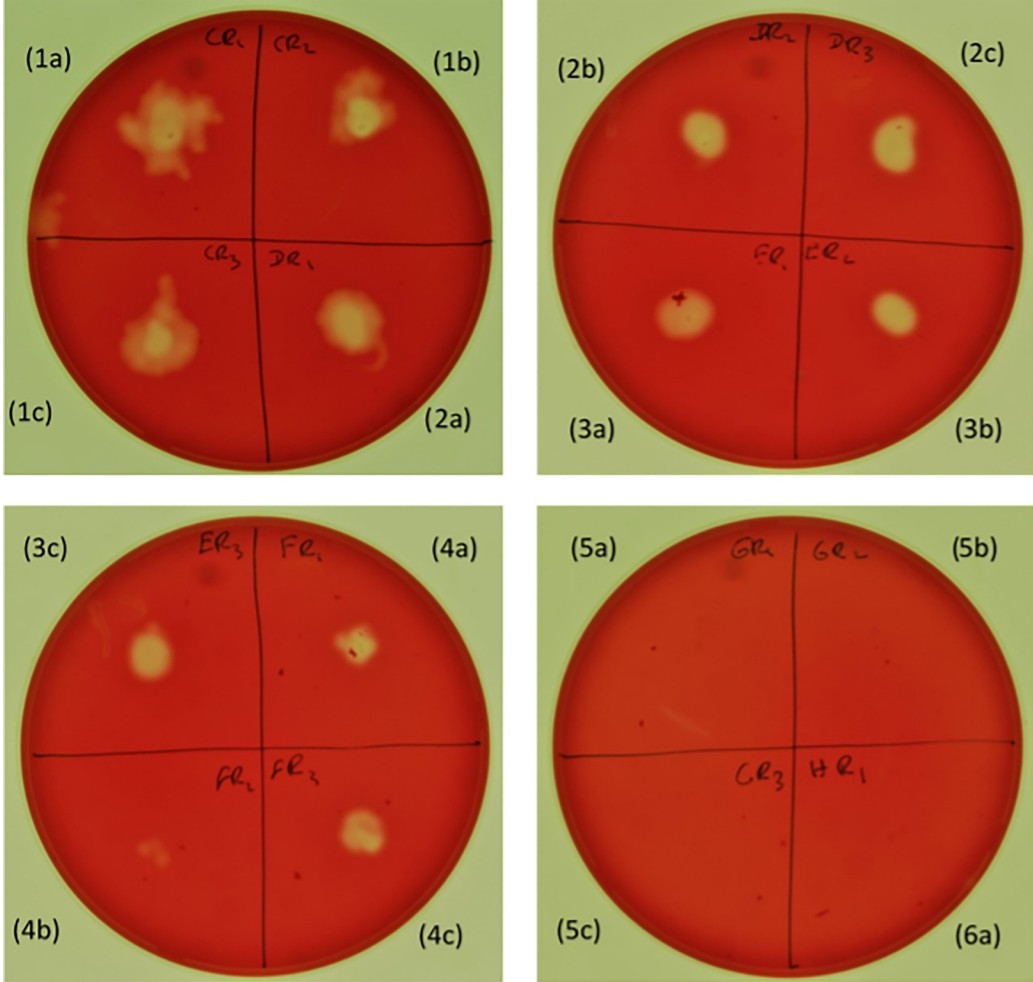

**Fig 4. Visible growth of *B. hampsonii* on trypticase soy agar (TSA) with different starting inoculum sizes.** Three replicates were included for each inoculum and a total volume of 2 µl were spotted onto each plate. (1a-1c) 8,000 CFU/spot, (2a-2c) 800 CFU/spot, (3a-3c) 80 CFU/spot, (4a-4c) 8 CFU/spot, (5a-5c) >1 CFU/spot, (6a) >1 CFU/spot. Haemolysis was used as an indicator of visible growth.

*hampsonii* isolate. There were 8 isolates, including representatives from each recognized species with lincomycin MICs of $\geq$128 µg/ml.

Except for *B. pilosicoli*, all isolates displayed low chloramphenicol MICs ($\leq$ 16 µg/ml) with an $MIC_{50}$ of 2 µg/ml. Similarly, tetracycline MICs were low with an $MIC_{50}$ of $\leq$ 0.25 µg/ml for all species except *B. pilosicoli* where an $MIC_{50}$ of 0.5 µg/ml. The MICs of nalidixic acid were nearly uniformly high; over the entire collection an $MIC_{50}$ of $\geq$ 64 µg/ml and $MIC_{90} \geq$ 128 µg/

**Table 1. Minimum inoculum of three *Brachyspira* species required for growth on liquid and solid media.**

| Species | Minimum inoculum size (CFU) yielding visible growth in liquid media | Minimum inoculum size (CFU) yielding haemolysis on agar |
|---|---|---|
| *B. pilosicoli* | $8.7 \times 10^5$ | 1 |
| *B. hyodysenteriae* | $5.4 \times 10^7$ | 11 |
| *B. hampsonii* | $4.1 \times 10^6$ | 8 |

**Table 2. Effect of inoculum size on observed minimum inhibitory concentration.**

| Inoculum dilutions | Minimum Inhibitory Concentration (MIC) (µg/ml) | | | | | | | | | | | |
|---|---|---|---|---|---|---|---|---|---|---|---|---|
| | Tiamulin | | | Tylosin | | | Ampicillin | | | Nalidixic acid | | |
| | R-1 | R-2 | R-3 | R-1 | R-2 | R-3 | R-1 | R-2 | R-3 | R-1 | R-2 | R-3 |
| *B. hyodysenteriae* (G44) | | | | | | | | | | | | |
| 0:0, $1.1\times10^9$ | 8 | 8 | 8 | 16 | 16 | 16 | >128 | >128 | >128 | >128 | >128 | >128 |
| 1:5, $2.2\times10^8$ | 8 | 8 | 8 | 8 | 8 | 8 | >128 | >128 | >128 | >128 | >128 | >128 |
| 1:25, $4.4\times10^7$ | 8 | 8 | 8 | 8 | 8 | 8 | >128 | >128 | >128 | >128 | >128 | >128 |
| 1:125, $8.8\times10^6$ | 8 | 8 | 4 | 8 | 8 | 4 | 4 | 32 | 64 | 4 | 4 | 64 |
| 1:625, $1.8\times10^6$ | 4 | 4 | 8 | 2 | 4 | 8 | 1 | 1 | 4 | 0.5 | ≤0.25 | ≤0.25 |
| 1:3125, $3.5\times10^5$ | 1 | 1 | 2 | 1 | 4 | 4 | 1 | 1 | 0.5 | ≤0.25 | ≤0.25 | ≤0.25 |
| *B. hampsonii (30446)* | | | | | | | | | | | | |
| 0:0, $3.6\times10^8$ | 1 | 1 | 1 | >128 | >128 | >128 | >128 | >128 | >128 | >128 | >128 | >128 |
| 1:5, $7.2\times10^7$ | 0.5 | 0.5 | 0.5 | 64 | 64 | 64 | 8 | 8 | 8 | 16 | 16 | 16 |
| 1:25, $1.4\times10^7$ | ≤0.25 | ≤0.25 | 0.5 | 8 | 8 | 4 | ≤0.25 | ≤0.25 | 0.5 | 2 | 2 | 4 |
| 1:125, $2.4\times10^6$ | ≤0.25 | ≤0.25 | ≤0.25 | 2 | 2 | 1 | ≤0.25 | ≤0.25 | 0.5 | 2 | 2 | 1 |
| 1:625, $5.7\times10^5$ | ≤0.25 | ≤0.25 | ≤0.25 | 1 | 1 | ≤0.25 | ≤0.25 | ≤0.25 | ≤0.25 | ≤0.25 | ≤0.25 | 0.5 |
| 1:3125, $1.1\times10^5$ | NGC[a] | NGC | NGC | NGC | NGC | NGC | NGC | NGC | NGC | NGC | NGC | NGC |
| *B. pilosicoli* | | | | | | | | | | | | |
| 0:0, $9\times10^8$ | ≤0.25 | ≤0.25 | ≤0.25 | 8 | 8 | 8 | 2 | 2 | 2 | 32 | 32 | 32 |
| 1:5, $1.8\times10^8$ | ≤0.25 | ≤0.25 | ≤0.25 | 4 | 4 | 4 | 2 | 2 | 2 | 8 | 8 | 8 |
| 1:25, $3.6\times10^7$ | ≤0.25 | ≤0.25 | ≤0.25 | 4 | 4 | 4 | 1 | 1 | 1 | 4 | 4 | 4 |
| 1:125, $7.2\times10^6$ | ≤0.25 | ≤0.25 | ≤0.25 | 1 | 0.5 | 0.5 | 0.5 | 0.5 | ≤0.25 | ≤0.25 | 4 | 2 |
| 1:625, $1.2\times10^6$ | ≤0.25 | ≤0.25 | ≤0.25 | ≤0.25 | ≤0.25 | 0.5 | 0.5 | ≤0.25 | ≤0.25 | ≤0.25 | ≤0.25 | 2 |
| 1:3125, $2.5\times10^5$ | NGC | NGC | NGC | NGC | NGC | NGC | NGC | NGC | NGC | NGC | NGC | NGC |

For each antimicrobial and starting inoculum size, three replicates were tested. Cells above the thick horizontal line indicate inoculum concentrations where consistent MICs were observed for all three replicates. NGC = no growth in control plates, R-1, R-2, R-3 = replicates 1 to 3.

ml were found. Finally, a wide distribution of ampicillin and amoxicillin and clavulanic acid MICs were observed among *B. hampsonii*, *B. hyodysenteriae* and *B. pilosicoli*, with an overall $MIC_{90}$ of >128 µg/ml. with MICs across the range of concentrations tested were identified. Conversely, *B. murdochii*, *B. innocens* were found to have ampicillin and amoxicillin + clavulanic acid MICs at the low end of the MIC range with an $MIC_{50}$ of ≤ 0.25 µg/ml.

## Discussion

The current lack of standardized antimicrobial susceptibility tests for *Brachyspira* species is an important diagnostic constraint which limits evidence-based application therapeutic guidance, and the detection of emerging resistance. The development of a standardized susceptibility test method is therefore crucial for the control of *Brachyspira*-associated disease. Previous studies have failed to describe a standard protocol for conducting antimicrobial susceptibility testing for *Brachyspira*, including a standardized starting inoculum [11, 17, 34]. The use of inocula varying by up to two orders of magnitude in previous investigations (from $1 \times 10^5–5 \times 10^5$ CFU/ml for broth dilution and $1 \times 10^4–1 \times 10^6$ CFU/spot for agar dilution) likely affects the MICs observed in those studies [11, 35]. This inconsistency makes it impossible to reliably compare data between laboratories. In this study we determined that there is minimum inoculum size require to obtain a visible growth, and that observed MIC is also affected by the starting bacterial concentration. Although the impact of inoculum density on MIC is well-

**Table 3. Reproducibility of the standard agar dilution method developed.**

| Drug | Observations with 100% agreement %, (n) | One doubling dilution difference observation (± 1) %, (n) | Reproducible %, (n) | Number of observations with MIC≤0.25 µg/ml %, (n) | More than one doubling dilution difference observations %, (n) | Correlation (Kendall's tau-b) | The measure of agreement (Kappa) |
|---|---|---|---|---|---|---|---|
| TIA | 91, (n = 32) | 6, (n = 2) | 97, (n = 34) | 74, (n = 26) | 3 (n = 1) | 0.921 | 0.795 (*p = 0.000*) |
| VAL | 80, (n = 28) | 20, (n = 7) | 100, (n = 35) | 77, (n = 27) | 0 (n = 0) | 0.793 | 0.443 (*p = 0.000*) |
| TYL | 63, (n = 22) | 17, (n = 6) | 80, (n = 28) | 3, (n = 1) | 20 (n = 7) | 0.722 | 0.517 (*p = 0.000*) |
| TYV | 69, (n = 24) | 20, (n = 7) | 89, (n = 31) | 26, (n = 9) | 11 (n = 4) | 0.888 | 0.631 (*p = 0.000*) |
| LIN | 63, (n = 22) | 17, (n = 6) | 80, (n = 28) | 6, (n = 2) | 20 (n = 7) | 0.722 | 0.517 (*p = 0.000*) |
| TET | 71, (n = 25) | 23, (n = 8) | 94, (n = 33) | 49, (n = 17) | 6 (n = 2) | 0.835 | 0.575 (*p = 0.000*) |
| CHO | 80, (n = 28) | 17, (n = 6) | 97, (n = 34) | 0, (n = 0) | 3 (n = 1) | 0.816 | 0.694 (*p = 0.000*) |
| NAL | 48, (n = 17) | 46, (n = 16) | 94, (n = 33) | 0, (n = 0) | 6 (n = 2) | 0.549 | 0.297 (*p = 0.010*) |
| AMP | 80, (n = 28) | 14, (n = 5) | 94, (n = 33) | 26, (n = 9) | 6 (n = 2) | 0.913 | 0.774 (*p = 0.000*) |
| AUG | 69, (n = 24) | 23, (n = 8) | 92, (n = 32) | 29, (n = 10) | 8 (n = 3) | 0.880 | 0.606 (*p = 0.000*) |

Test results were considered to be reproducible if MICs between replicates were identical or differed by no more than a single doubling dilution. Total number of isolates = 35. The fifth column indicates the number of isolates inhibited beyond the lowest concentration tested; MIC ≤ 0.25 µg/ml. The first and second observations of MICs from each antimicrobial were statistically compared, and correlation and agreement measurements were listed in the last two columns respectively.

TIA = tiamulin, VAL = valnemulin, TYL = tylosin, TYV = tylvalosin, LIN = lincomycin, CHO = chloramphenicol, TET = tetracycline, NAL = nalidixic acid, AMP = ampicillin, AUG = amoxicillin + clavulanic acid.

recognized in other bacterial taxa, it has not been systematically investigated in *Brachyspira* spp. [35–39]. The development of a standard curve defining the relationship between colony count (hemolysis forming unit) and the optical density for *B. hampsonii*, *B. hyodysenteriae* and *B. pilosicoli* was a critical first step in our study. Optical density is a rapid method of determining the density of a bacterial culture which allows test inoculum to be standardized in susceptibility testing.

The results of this investigation confirmed that the starting inoculum was a limiting factor for the growth of *Brachyspira* spp. particularly in liquid media. *B. pilosicoli* required the lowest and *B. hyodysenteriae* the highest starting bacterial concentration to obtain growth on both media types. These results also demonstrate that viable broth cultures require a higher starting inoculum compared to cultures on agar. These observations highlight the recommendation to use the agar dilution method, which is suggested when testing other fastidious anerobic bacteria [40].

For agar dilution a concentration of $2–4 \times 10^5$ CFU/spot was chosen in this investigation. Standardizing our method using this concentration has the advantage of being consistent with CLSI guidelines and therefore being familiar to clinical diagnosticians, which should facilitate the incorporation of this assay methodology into diagnostic workflows. By using this method, this study demonstrated consistent results, both between replicates and on following repeated testing. This is in contrast to the poor reproducibility which has been reported for the broth micro-dilution test; uninterpretable results due to "skipped wells" (the well without growth, despite the occurrence of growth in wells with higher concentrations) are recognized when testing lincosamides and macrolides [17, 41]. Interestingly, a recent study aiming to validate the a broth microdilution test also reported an inadequately described means of standardizing the starting inoculum [17]. Although we report a high degree of test reproducibility in the current study, there were a number of bacterial isolates that were inhibited by the lowest concentration of drug tested 0.25 µg/ml. It is therefore possible that for isolates with very low MICs there was variability in test performance below the limit of detection of our assay.

Agar dilution antimicrobial susceptibility testing is not widely performed in clinical diagnostic laboratories. Because it is laborious to perform and requires the preparation of plates

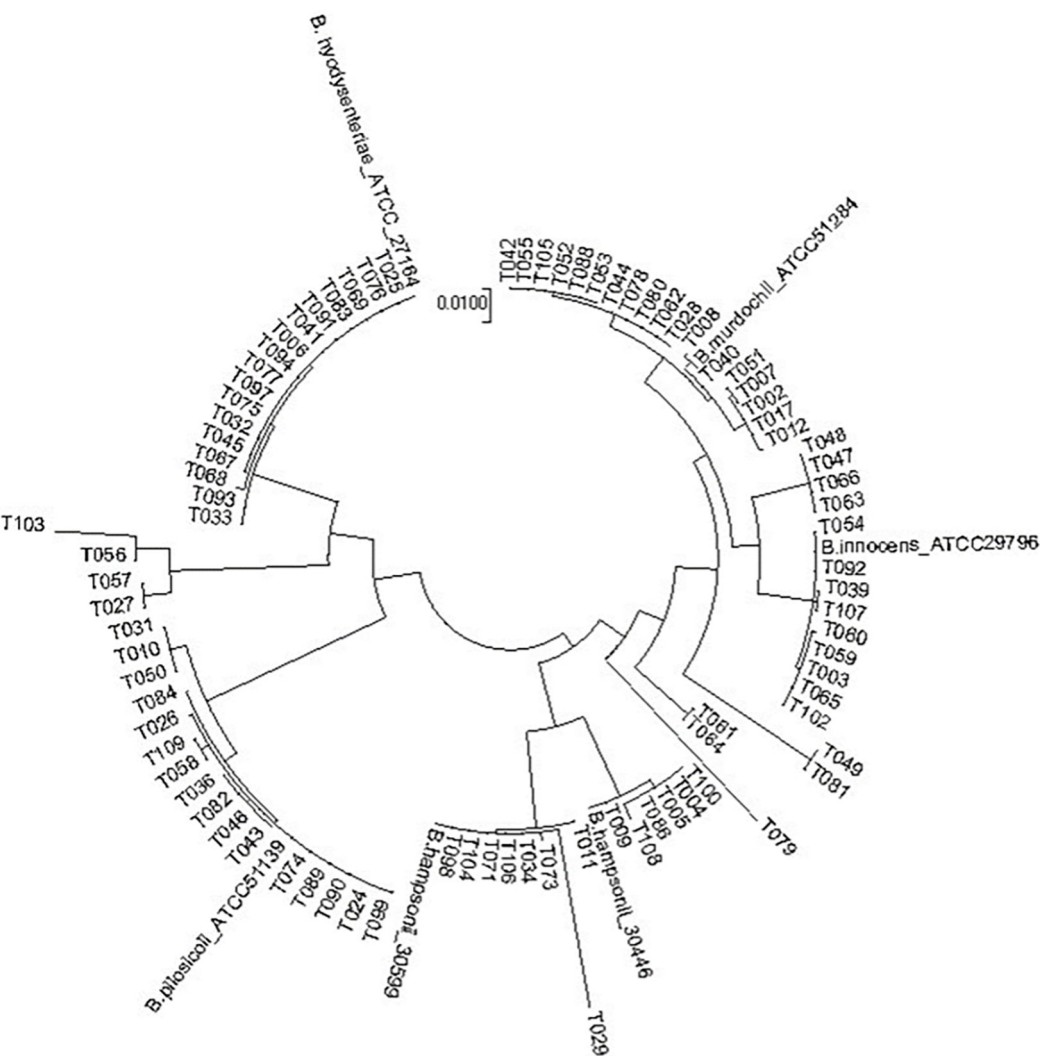

**Fig 5. Phylogenetic tree of *Brachyspira* spp. tested.** Phylogenetic tree of partial *nox* sequences of diagnostic isolates and types strains of *B. hyodysenteriae*, *B. pilosicoli*, *B. innocens* and *B. murdochii* based on a 765 base pair alignment. Sequences associated with *B. hamponii* genomevar I (*B. hampsonii*_30599; AOMM00000000), and *B. hampsonii* genomevar II (*B. hampsonii*_30446; IDAC No 161111–01, ALNZ00000000) are also included.

containing many different concentrations of antibiotics, it's use is limited to research studies and reference laboratories. However, for laboratories using culture-based methods for *Brachyspira* diagnostics it would be practical to implement a targeted susceptibility testing service for a smaller panel of key drugs using the method described here.

The MIC distribution of bacterial populations can be classified as homogenous, bi-modal, or multi-modal [42]. This classification may provide clues to the mechanism of resistance acquisition. When bacteria acquire a resistance gene for example, there may be a distinct change in resistance leading to a bi-modal MIC distribution separating wildtype and resistant organisms into distinct populations. Conversely, step-wise MIC increases may be seen with the acquisition of successive resistance conferring mutations, as is observed with fluoroquinolone MICs following successive topoisomerase gene mutations [42]. Although the isolates tested in this study were conveniently collected from diagnostic submissions, they came from

**Table 4. The number of isolates from each production system.**

| Production system | Total number of isolates | Number of each species | | | | | |
|---|---|---|---|---|---|---|---|
| | | BH | BHM | BP | BM | BI | NC |
| A | 3 | | | | 2 | | 1 |
| B | 39 | 15 | 8 | 3 | 4 | 6 | 3 |
| C | 16 | | 2 | | 8 | | 6 |
| D | 5 | | 2 | 1 | | | |
| E | 13 | 1 | | 6 | 3 | | 3 |
| F | 5 | | | 4 | 1 | | |
| G | 4 | 1 | | 2 | | 1 | |
| H | 2 | | | 2 | | | |

BH = *B. hyodysenteriae* (n = 17), BHM = *B. hampsonii* (n = 14), BP = *B. pilosicoli* (n = 16), BM = *B. murdochii* (n = 18), BI = *B. innocens* (n = 7), NC = non-clustering (n = 13).

eight epidemiologically distinct productions; this diverse strain origin makes the observed MIC distributions more informative than if the isolates were epidemiologically linked.

Anecdotally, tiamulin is the most used drug to treat *Brachyspira*-associated diseases in western Canada. In the Czech Republic, one study found that the MICs to tiamulin and valnemulin increased significantly between the periods 1997–98 and 1999–2001 among a collection of *B. hyodysenteriae* isolates [43]. Similar findings were reported in a Japanese study among *B. hyodysenteriae* isolates between 1985 and 2009 [44]. Although the lack of a standardized methodology prevents comparing MIC results between labs, these longitudinal studies were conducted within single laboratories (where a consistent methodology would have been employed), suggesting that pleuromutilin MICs are increasing [44]. In contrast, isolates originating from the United States were observed to have lower pleuromutilin MICs [24]. Perhaps not surprisingly, our results suggest that the situation in western Canada is more similar to the United States where *B. hyodysenteriae* and *B. hampsonii* isolates have very low MICs ($\leq 1$ µg/ml) to pleuromutilin drugs. One study from the United States categorized isolates with MICs >8 µg/ml as having "decreased susceptibility" or "resistance" to tiamulin [24]. In our study, 44% *B. pilosicoli* isolates had high pleuromutilin MICs (>8 µg/ml), possibly indicating resistance. Consistent with our observations, previous studies have demonstrated the tendency of *B. pilosicoli* to develop resistance to macrolides, lincosamides and pleuromutilins more rapidly than other species [24, 45]. Furthermore, cross-resistance between valnemulin and tiamulin in *B. pilosicoli* is a likely explanation for the similar MIC distributions we observed for both tiamulin and valnemulin in *B. pilosicoli* [46].

A recent study from the United States described high MICs to lincomycin ($MIC_{50}$ = 8 µg/ml, $MIC_{90}$ = 32 µg/ml) and tylosin ($MIC_{50}$>128 µg/ml, $MIC_{90}$>128 µg/ml) among *Brachyspira* isolates [23]. Interestingly, in our study the MIC distributions observed were heterogeneous, with MICs across the spectrum tested for both antimicrobials. Previous studies have identified single nucleotide polymorphisms in the 23S ribosomal RNA gene and the ribosomal protein (L3), as well as the acquisition of the *lnu*C and *tva*A that are associated with resistance to protein synthesis inhibitors [47–50]. Those studies observed bimodal distribution of MICs to tylosin in both *B. hyodysenteriae* and *B. pilosicoli*, with isolates possessing 23S rRNA single nucleotide polymorphisms having higher MICs than those isolates without SNPs [51, 52]. In the current investigation, *B. pilosicoli* was observed to have had a bimodal MIC distribution for tylosin and the highest $MIC_{50}$ (>128 µg/ml) compared to other species. Similar to the pleuromutilins, It has been suggested that the recombinant population structure and the

**Table 5. Antimicrobial minimum inhibitory concentration distribution.**

| Antimicrobial | Species[a] | ≤0.25 | 0.5 | 1 | 2 | 4 | 8 | 16 | 32 | 64 | 128 | >128 | MIC$_{50}$ | MIC$_{90}$ |
|---|---|---|---|---|---|---|---|---|---|---|---|---|---|---|
| Tiamulin | BHM | 12 | | 2 | | | | | | | | | ≤0.25 | 1 |
| | BH | 16 | | | 1 | | | | | | | | ≤0.25 | ≤0.25 |
| | BP | 5 | 2 | 2 | | | 2 | 1 | 2 | 2 | | | 1 | 64 |
| | BM | 15 | | 1 | | | 2 | | | | | | ≤0.25 | 8 |
| | BI | 9 | | | | | | | | | | | ≤0.25 | ≤0.25 |
| | NC | 13 | | | | | | | | | | | ≤0.25 | ≤0.25 |
| Valnemulin | BHM | 14 | | | | | | | | | | | ≤0.25 | ≤0.25 |
| | BH | 17 | | | | | | | | | | | ≤0.25 | ≤0.25 |
| | BP | 9 | 1 | | 2 | 1 | | | 2 | 1 | | | ≤0.25 | 32 |
| | BM | 16 | 1 | | 1 | | | | | | | | ≤0.25 | 0.5 |
| | BI | 9 | | | | | | | | | | | ≤0.25 | ≤0.25 |
| | NC | 13 | | | | | | | | | | | ≤0.25 | ≤0.25 |
| Tylosin | BHM | 3 | | 2 | 1 | 2 | 4 | | | | | 2 | 4 | >128 |
| | BH | 2 | | 1 | 1 | 4 | 3 | 3 | 1 | | | 2 | 8 | >128 |
| | BP | | | | | 5 | 1 | | | | 1 | 9 | >128 | >128 |
| | BM | 3 | | | | 6 | 4 | 1 | | | | 4 | 4 | >128 |
| | BI | | | 1 | 3 | 2 | 1 | | | | | 2 | 4 | >128 |
| | NC | 3 | | 1 | 3 | 1 | | 1 | | 1 | | 3 | 4 | >128 |
| Tylvalosin | BHM | 8 | | 1 | 2 | 1 | | 1 | | | | 1 | ≤0.25 | 16 |
| | BH | 7 | 2 | 4 | 2 | 1 | | 1 | | | | | 0.5 | 4 |
| | BP | 1 | 4 | | 1 | 4 | 2 | | | 1 | | 3 | 2 | >128 |
| | BM | 4 | 7 | 2 | 1 | 1 | | 1 | | 1 | 1 | | 0.5 | 64 |
| | BI | 6 | | 1 | | | | 1 | | | | 1 | ≤0.25 | >128 |
| | NC | 8 | 1 | 2 | | 1 | 1 | | | | | | ≤0.25 | 4 |
| Lincomycin | BHM | 6 | 1 | 3 | | 1 | | | | 1 | | 2 | 0.5 | >128 |
| | BH | 6 | 4 | 4 | 1 | 1 | | | | | 1 | | 0.5 | 4 |
| | BP | 1 | | 2 | 2 | 2 | | | 2 | 4 | 2 | 1 | 32 | 128 |
| | BM | 5 | 1 | 2 | 6 | | | | 2 | 1 | 1 | | 2 | 64 |
| | BI | 4 | 1 | 1 | | 1 | | | | 1 | 1 | | 0.5 | 128 |
| | NC | 1 | 1 | 6 | | | | 2 | | 3 | | | 1 | 64 |
| Chloramphenicol | BHM | | 1 | 4 | 7 | 1 | | 1 | | | | | 2 | 4 |
| | BH | 2 | | 2 | 7 | 3 | 3 | | | | | | 2 | 8 |
| | BP | | | 1 | 6 | 5 | 2 | 2 | | | | | 4 | 16 |
| | BM | 2 | | 2 | 1 | 3 | | | | | | | 2 | 4 |
| | BI | | 1 | 1 | 6 | 1 | | | | | | | 2 | 4 |
| | NC | | 1 | 4 | 7 | 1 | | | | | | | 2 | 2 |
| Tetracycline | BHM | 8 | 1 | 3 | | | 1 | 1 | | | | | ≤0.25 | 8 |
| | BH | 9 | 4 | 2 | 1 | 1 | | | | | | | ≤0.25 | 2 |
| | BP | 7 | 3 | 3 | 1 | 1 | 1 | | | | | | 0.5 | 4 |
| | BM | 13 | 2 | 1 | 2 | | | | | | | | ≤0.25 | 2 |
| | BI | 6 | | 1 | 2 | | | | | | | | ≤0.25 | 2 |
| | NC | 10 | 1 | | 2 | | | | | | | | ≤0.25 | 2 |
| Nalidixic acid | BHM | 1 | | | 1 | | 1 | | 1 | 1 | 4 | 5 | 128 | >128 |
| | BH | | | | | | | | 3 | 4 | 5 | 5 | 128 | >128 |
| | BP | | | | | | | | 3 | | 9 | 4 | 128 | >128 |
| | BM | 1 | | | | | | | 7 | 4 | 6 | | 64 | 128 |
| | BI | | | | | | | | 1 | | 7 | 1 | 128 | >128 |
| | NC | | | | | | | 1 | 1 | 6 | 5 | | 64 | 128 |

(*Continued*)

**Table 5.** (Continued)

| Antimicrobial | Species[a] | ≤0.25 | 0.5 | 1 | 2 | 4 | 8 | 16 | 32 | 64 | 128 | >128 | MIC$_{50}$ | MIC$_{90}$ |
|---|---|---|---|---|---|---|---|---|---|---|---|---|---|---|
| Ampicillin | BHM | 3 | 1 | 2 | | | | 1 | | | | 4 | 32 | >128 |
| | BH | 3 | | 4 | | | 2 | | 1 | | | 7 | 16 | >128 |
| | BP | | 2 | 5 | | | 1 | | 1 | | | 7 | 8 | >128 |
| | BM | 15 | 2 | | | | | | | | | 1 | ≤0.25 | 0.5 |
| | BI | 4 | 4 | | | | | | | | | 1 | ≤0.25 | 0.5 |
| | NC | 11 | 2 | | | | | | | | | | ≤0.25 | 0.5 |
| Amoxicillin + clavulanic acid | BHM | 3 | 1 | 2 | | | | | 1 | | | 7 | 32 | >128 |
| | BH | 1 | 4 | | 1 | | | 1 | 2 | | 1 | 7 | 32 | >128 |
| | BP | 1 | | 2 | 1 | 2 | 3 | | 1 | | 2 | 4 | 8 | >128 |
| | BM | 17 | | | | | | | | | | 1 | ≤0.25 | ≤0.25 |
| | BI | 8 | | | | | | | | | | 1 | ≤0.25 | >128 |
| | NC | 13 | | | | | | | | | | | ≤0.25 | ≤0.25 |

[a]BHM = *B. hampsonii*, BH = *B. hyodysenteriae*, BP = *B. pilosicoli*, BM = *B. murdochii*, BI = *B. innocens*, NC = non-clustering. In columns 3–13, the number of isolates were inhibited at each concentration. The MIC$_{50}$ and MIC$_{90}$, the concentrations at which 50% and 90% of the isolates are inhibited respectively, are in the final which were calculated based on the MICs distributions.

substantial amount of genomic variation in *B. pilosicoli* may contribute to the emergence of antimicrobial resistance [24, 53].

Interestingly, MICs for tylvalosin tended to be elevated in isolates with high tylosin and lincomycin MICs (Table 5). All 6 isolates which were uninhibited by tylvalosin (>128 μg/ml) had tylosin MICs > 128 μg/ml, and lincomycin MICs ≥ 32 μg/ml. Macrolides, lincosamides, and pleuromutilins bind to the peptidyl transferase center (PTC) of 23S rRNA of the 50S ribosome and prevent the peptide bond formation and thereby prevent the protein synthesis of bacteria [51]. Nucleotide mutations or methylations which occur in the highly conserved main loop of domain V in PTC have been shown to lead to resistance to macrolides, lincosamides, streptogramins and pleuromutilins [47]. These mutations have been previously studied among *Brachyspira* spp. and their relationship with observed MIC has been documented [22, 54]. The common target of lincomycin, tylosin and tylvalosin (the 23S rRNA) may be responsible for the cross resistance observed in our study.

Although chloramphenicol is banned in food animals in Canada, it is commonly included in antimicrobial resistance surveillance programs [55, 56]. In the current study, all isolates were inhibited by 16 μg/ml. The fluoroquinolones, used in both veterinary and human medicine, are also frequently included in resistance surveillance programs. Resistance to these drugs can occur by target modifications, decreased permeability, efflux and target protection [54, 57]. There is evidence of intrinsic quinolone resistance among Gram-negative anaerobes, although it has not been determined if this is the case for *Brachyspira* spp. [58]. Most of the isolates tested in this study among all species had very high nalidixic acid MICs (>16 μg/ml), possibly indicating intrinsic resistance.

High MICs to ampicillin and amoxicillin + clavulanic acid was observed in *B. hyodysenteriae*, *B. hampsonii* and *B. pilosicoli*. While β-lactams are not used to treat *Brachyspira*-associated infections, the use of penicillin for treating other infections in pigs may have contributed to the selection of resistance to these drugs in *Brachyspira* [59–61]. Previous studies found that *B. pilosicoli* with high β-lactam drugs MICs possessed the OXA-63 gene, a class D β-lactamase [59].

## Conclusions

In this study we developed a standard agar dilution method with high reproducibility in our laboratory. This method reduces the variability of the susceptibility test results and will allow results to be compared between laboratories. It was encouraging to find low pleuromutilin MICs for the most important pathogens *B. hampsonii* and *B. hyodysenteriae* while signals of emerging resistance were detected among *B. pilosicoli*. The results of this study emphasize the importance of diagnostic testing for the identification of *Brachyspira* species and for therapeutic selection. Continued monitoring and of the susceptibility of isolates is warranted to detect emerging resistance. Finally, this study highlights the persistent challenge of a lack of standardized set of interpretive criteria to categorize *Brachyspira* isolates as susceptible or resistant. This is a topic that rbabequires additional investigation for the development of evidence-based, clinically predictive criteria.

## Supporting information

**S1 Table. Optical density and culture density measurements used in development of standard curve.**
(DOCX)

## Acknowledgments

We would like to thank Michelle Sniatynski, Champika Fernando and Drs. Janet Hill, Sarah Parker and Sheryl Gow for assistance with this study. Finally, we thank Eco Animal Health, United Kingdom for their gift of tylvalosin powder used in testing.

## Author Contributions

**Conceptualization:** D. G. R. S. Kulathunga, Joseph E. Rubin.

**Data curation:** D. G. R. S. Kulathunga.

**Formal analysis:** Joseph E. Rubin.

**Funding acquisition:** Joseph E. Rubin.

**Investigation:** D. G. R. S. Kulathunga.

**Methodology:** Joseph E. Rubin.

**Project administration:** D. G. R. S. Kulathunga, Joseph E. Rubin.

**Supervision:** John C. S. Harding, Joseph E. Rubin.

**Validation:** D. G. R. S. Kulathunga.

**Writing – original draft:** D. G. R. S. Kulathunga, Joseph E. Rubin.

**Writing – review & editing:** D. G. R. S. Kulathunga, John C. S. Harding, Joseph E. Rubin.

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
