## [Decision Letter · Decision Letter 0]

15 Feb 2023

PONE-D-22-35189Antimicrobial susceptibility of Western Canadian Brachyspira isolates: Development and standardization of an agar dilution susceptibility test methodPLOS ONE

Dear Dr. Rubin,

Thank you for submitting your manuscript to PLOS ONE. After careful consideration, we feel that it has merit but does not fully meet PLOS ONE’s publication criteria as it currently stands. Therefore, we invite you to submit a revised version of the manuscript that addresses the points raised during the review process.

We look forward to receiving your revised manuscript.

Kind regards,

Marwa Ibrahim Abd El-Hamid

Academic Editor

PLOS ONE

Journal Requirements:

3. We noted in your submission details that a portion of your manuscript may have been presented or published elsewhere. Please clarify whether this publication was peer-reviewed and formally published. If this work was previously peer-reviewed and published, in the cover letter please provide the reason that this work does not constitute dual publication and should be included in the current manuscript.

Reviewers' comments:

Reviewer's Responses to Questions

**Comments to the Author**

1. Is the manuscript technically sound, and do the data support the conclusions?

Reviewer #1: Yes

Reviewer #2: Partly

Reviewer #3: Yes

2. Has the statistical analysis been performed appropriately and rigorously? 

Reviewer #1: Yes

Reviewer #2: I Don't Know

Reviewer #3: I Don't Know

3. Have the authors made all data underlying the findings in their manuscript fully available?

Reviewer #1: Yes

Reviewer #2: Yes

Reviewer #3: Yes

4. Is the manuscript presented in an intelligible fashion and written in standard English?

Reviewer #1: Yes

Reviewer #2: No

Reviewer #3: Yes

5. Review Comments to the Author

Reviewer #1: The manuscript by Rubin et al. describes a well-designed study for a species where standardized methods are lacking. The results are clearly presented, and my major comments are on the data analysis.

Line 94: A proper EUCAST reference for this would be the EUCAST Breakpoint Tables (in which breakpoints for Brachyspira spp. are lacking).

Line 110: Please explain “BJ agar” when first mentioned.

Lines 246-247: Since several MICs were below the lowest concentration of the dilution series, I suggest adding information on the positive control. Was there always haemolysis in the positive control for MICs read as ≤ the lowest value? Also, were MIC endpoints difficult to define according to the established criteria? If so, were MIC endpoints read by more than one technician?

Lines 260-263, 435-441 and Table 3: The authors explain that MICs were regarded to be in complete agreement when MICs were identical or within one dilution. In Table 3, there is a column showing how many results that were below the lowest concentration of the dilution series, but I lack a comment on how this affected the calculations of reproducibility. Also, its should me mentioned as a limitation of the study that many MICs were below the lowest concentration, which makes it difficult to assess the reproducibility.

Discussion: The authors show good reproducibility of a new standardized methodology for agar dilution of Brachyspira spp., but it is not discussed how/if this method can be implemented in other laboratories. Please add some text on if you consider that this is a method for a reference laboratory or if it is something that can be implemented/used in routine laboratories working with Brachyspira spp.

Reviewer #2: Dear authors,

I recommend “Major revision” on this article

I would like to congratulate you for the great amount of work you performed; but unfortunately the report you gave here is very confusing, too long then I forgot very fast what was the goal of the study. Maybe you should split the paper in at least 2 : 1/ with the new methodology you offer to standardize 2/ the clinical results that you could compare with the use, clinical output, etc..

for the rest of my review, I will focus on the idea that you write a paper for the methodology of susceptibility testing of Brachyspira.

1/ what does standardization mean for you ? is a method standardized if you perform (even with hundreds of repetitions) in only one center ? I guess not

Please explore documents for EUCAST or CLSI to determine what your definition of a standardized method is. I especially recommend to study document M23 5th ED 2018 from CLSI which is free.

2/there are plenty of interesting documents for the purpose of your study at CLSI but unfortunately I think you did not pick the right one. M23 is the most important with M11. You cite plenty of documents but not always the right one at the right time. Did you notice that reference 34= reference 19, it is the same document. Line 234 : I am very surprised that CLSI recommend TSA agar plate : I thought it was always a Muller hinton (MH) base ? you cite reference 12, document M100 as reference but to my knowledge M100 is supposed to provide interpretation criteria for bacteria of human clinical importance that grow aerobically… I am lost… By the way, make sure you use up-to-date documents, the M100 you cite is 5 editions late and then totally expired if you do not justify why you use this old document

3/it is very complicated to cultivate Brachyspira, so why do you wish to perform antibiogram sensu lato ? I am not sure I understood the clue of such a complicated method. Is there issues on the pig health side? therapeutic failure? Is it just for research and epidemiological purposes ? Is it an issue for human health? it is not clear to me why such a method should be implemented. It seems silly but it is not. If you define what kind of output for public you are looking for you can decide which kind of breakpoint you want to set. See CLSI document M23 chapiter 5. By reading you, I did not understand which type of breakpoint you want to set here and for which purposes. Again is this antibiogram helping vets to decide how they shall treat the sick pigs ?

Is there a justification to split you results at species level. I noticed that most of the other studies you cite is collapsing results at gender level ?

4/figures and tables are sometimes useless and confusing

What am I supposed to see on Figure5 ? Table 4 why do I need to know the origin of the strains? is it important for your conclusion ? is it finally clinical results ?

5/ your interpretations / conclusions are sometimes questionable

Line652-653: is it finally comparable or not. I am lost

Line668-669: I disagree with your “step-wise” hypothesis. It is also conceivable that you are facing a multitude of different mechanism.

My best advice: restrict your goal and explain it. Don’t forget to define the term you use (at least for you), your readers would follow your point. As it is it is too complicated to memorize your take home message ( which I haven’t been able to understand)

Reviewer #3: The present study ”Antimicrobial susceptibility of Western Canadian Brachyspira isolates: Development and standardization of an agar dilution susceptibility test method” by Kulathunga et al developed a standardized protocol for conducting agar dilution susceptibility testing of Brachyspira spp. and determine the susceptibility of 32 isolates from Western Canadian Brachyspira using the standardized methodology. Following are the specific comment regarding this manuscript.

1. Page 5, Any pericular reason behind selecting the “B. pilosicoli (ATCC 51139), B. hyodysenteriae (JXNI00000000) and B. hampsonii genomovar II (IDAC No 161111-01, ALNZ00000000” for Development of a standard curve over the other ATCC strain mentioned on Page 9 line 193 to 196?

2. As the test is being standardized for antimicrobial susceptibility testing did the author’s used any known resistance strain with known mutations? If No, why?

3. Please provide the data (may be supplementary) on serial dilutions of the bacterial culture’s vs OD at 600nm.

4. Page 6; Line 126-129: “Following incubation, a drop of each culture was examined under a phase-contrast microscope at 400 magnification to confirm the presence of live, motile spirochetes. The optical density (at 600 nm) of cultures was then measured to determine the bacterial concentration” Here the bacterial concentration means CFU per ml? If yes, please mention.

5. Page 6, line 119 – 121: “The relationship between OD600nm, CFU/ml and genome equivalents/ml as measured by qRT PCR was previously determined and found to be consistent” Please provide the reference.

6. Why broth culture was incubated at 39 oC and agar plates are at 42 oC.

7. Page 7, line 133 - In the case of broth, visible turbidity compared to an uninoculated control was considered positive (growth….”. Did the authors examined the absence or absence of bacterial growth under a phase-contrast microscope to confirm the bacterial growth?

8. Why different dilution series were used for Development of a standard curve (1:1.1, 1:1.2, 1:1.3 and 1:2-1:512) and Determination of the minimum inoculum required to start a culture (1:10 dilution series (10-1 to 10-9)?

9. Why two different agar plates (BJ agar and Trypticase soy agar (TSA) + 5% sheep’s blood) were used during the experiments.

10. Page 10 - 11. Porcine clinical isolates identification: Please provide the details on how the phylogenetic tree was constructed (nucleotide substitution mode and bootstrap replicates)?

11. Figure 5: Please include the sequence from B. hyodysenteriae (JXNI00000000), B. hampsonii genomovar II (IDAC No 161111-01, ALNZ00000000) and also available sequence from different species (B. hyodysenteriae, B. hampsonii, B. pilosicoli, B. murdochii, B. innocens,) from Western Canadian region for phylogenetic analysis.

12. How did the authors confirms that the non-clustering (n=13) are from Brachyspira species?

13. Page 11 – 12: “When turbidity was observed, ODs were measured and bacterial density was adjusted to 1-2 X108 CFU/ml….” Please explain why?

14. Page 13: Development of an equation relating organism concentration to an optical density: As each Brachyspira species showed different CFU in liquid media and on agar media. Why the Avg equation were considered to calculate the CFU/mL and not the Brachyspira species specific equation.

15. Fig 2: Why the data were collected up to OD value 1 for B. pilosicoli and B.hampsonii,

16. Fig. 2. Why data were collected for OD values 0.1 to 0.4 for hyodysenteriae.

17. The information on the growth kinetics data on three Brachyspira species will provide the information on growth cycle. Did the authors study the growth kinetic for the three isolates? If yes, please provide the data (may be as supplementary).

18. Did the authors examined the culture microscopically for degree of clumping/aggregation and presence of dead bacteria before taking the OD values?

19. Did the authors confirmed the presence of known antibiotic resistance mutations for the isolates to confirm the resistance to antibiotics? If No, please explain why?

20. Did the authors submit the “nox gene” sequence data generated to GenBank? If no, please submit and provide GenBank accession number.

6. PLOS authors have the option to publish the peer review history of their article (what does this mean?). If published, this will include your full peer review and any attached files.

Reviewer #1: No

Reviewer #2: No

Reviewer #3: No

---

## [Author Response · Author response to Decision Letter 0]

25 Apr 2023

General comments:

Change in authors listed: In the initial submission, we had mistakenly included Dr. S.P. Gow in the manuscript file. Dr. Gow was not included as an author within the PLoS submission system and should not have been listed on the manuscript file. I apologize for this oversight.

The references have also been corrected as per the reviewer's suggestions. 

Reviewer 1

Line 94 - A reference to the EUCAST breakpoint tables has been added as suggested.

Line 110 - Thank you for this comment, a description of this media has been added as suggested.

Lines 246-247 - Some modifications and additional information were added to this section of the paper to clarify. Antimicrobial free (positive control) plates were included with each batch of plates; growth was always observed on these plates. Had growth not been observed, the experiment would have been repeated. Regarding test endpoints, we defined the MIC as the lowest concentration where no hemolysis was observed; in this study we used hemolysis as an indicator of growth as opposed to a colony or turbidity which would be done for the majority of organisms encountered by the clinical diagnostic laboratory. Using hemolysis as an indicator of growth is an established method when working with Brachyspira. Finally, endpoints were only read by DGRS Kulathunga who performed these experiments during her PhD. Although, this isn't something that I can report in the paper, these observations were only made following a period of extensive training so the researcher was not a novice in the field when these experiments were conducted.

Lines 260-263, 435-441 and Table 3 - Thank you for this comment. We struggled with how best to present this data; to be fully transparent with our date we included a column describing the number of observations with MIC ≤ 0.25 µg/ml. A sentence was added just after line 260 to explain that repeated observations ≤ 0.25 µg/ml were considered reproducible. In the discussion section lines 649-652 a sentence was added to address the limitation in interpreting reproducibility for isolates with MICs ≤ 0.25 µg/ml.

Discussion - Thank you for raising this point. We have added a recommendation for how we might see this method implemented into a clinical diagnostic laboratory on lines 653-658.

Reviewer 2:

Thank you for the comment recognizing the volume of work reflected in this manuscript. We agree that there is a lot of data here and that this manuscript includes 2 large bodies of work (1. method development and 2. testing isolate collection). We had lengthy discussions on whether to lump these data together and struggled with what would be the best approach, in the end we felt that for the benefit of the Brachyspira literature it would be better to present this data as a single body of work. We felt that this manuscript was much stronger if both parts were combined; the length of the paper and the number of figures we wanted to include factored into our decision to submit to PLoS One. 

Point 1 - We have added a description of what test factors are standardized according to the CLSI and EUCAST on lines 95-98.

Point 2 - Thank you for these detailed comments. 

• We have reviewed which CLSI documents are cited throughout the manuscript and made changed which documents are cited where necessary.

• Regarding line 234 - you are correct, the CLSI does not recommend TSA. This sentence has been modified to emphasize that the media was prepared according to the CLSI guidelines (procedure for measuring antimicrobial powders etc.) but that the antimicrobials were then incorporated into TSA vs. Mueller-Hinton agar.

• We have also modified the CLSI reference at line 234 - this should not in fact be M100 but the M07 document which described the procedure for agar dilution testing.

Point 3 - We absolutely appreciate that the method described here is somewhat complicated, and as an agar dilution protocol it is certainly more laborious than other methods such as disc diffusion or commercially prepared broth microdilution. However, we believe that we have demonstrated that the current method, unlike others previously reported, is sufficiently reliable to be considered as a tool in clinical diagnostic laboratories. We fully accept your criticism of 'what to do with the data', we are not yet at a point where we have validated clinical breakpoints as we do with other veterinary pathogens, so caution will be required when interpreting this data. This study is a necessary first step, before interpretive criteria can be developed, reliable and standardized test methodologies need to be developed. Ultimately, high quality antimicrobial susceptibility data is a cornerstone of diagnostic bacteriology; it is required to guide therapeutic selection, promote antimicrobial stewardship to reduce the selection pressure for resistance and finally to detect the emergence of antimicrobial resistance. 

With respect to stratifying data by species group, this was done for two primary reasons. First, we wanted to ensure that the most granular data possible was presented to allow researchers to re-interpret our data in the future. Second, because differences in intrinsic resistance between species are well recognized (see EUCAST Expected Resistant Phenotypes v1.2 January 2023 table for examples), we wanted to ensure that any such phenomenon was not masked by aggregating data within a single group. 

Point 4 

• Figure 5 - A figure legend has been added to explain this figure.

• Table 4 - A sentence has been added (lines 677-680). The main reason that it is important to document that the isolates in this study were from epidemiologically distinct production systems is to be transparent about how biased (homogeneous) our isolate collection is. If all isolates originated from the same farm it would not have been reasonable to discuss MIC distributions because the isolates would have been subjected to the same antimicrobial selection pressures.

Point 5

• Line 652-653 - A clarification has been added here. We want to highlight the fact that while methodological differences preclude comparisons between labs, results generated over time from within a single lab may be compared with less uncertainty.

• Line 668-669 - We have decided to delete this sentence in accordance with your suggestion.

Reviewer 3:

Point 1

• These three isolates were selected because they represent the three most commonly identified Brachyspira species causing infections in pigs in our region. Clarification was added to explain that these isolates were selected for this reason.

Point 2 

• In this study we did not include isolates posessing particular resistance genes or resistance conferring mutations. Because the isolate collection we tested was a set of diagnostic isolates from our institution, this information was not available. Before doing this study we were in the difficult position of having neither genotypic resistance data nor phenotypic susceptibility results for our collection which is why we performed our assays in triplicate and took other measures as described to be confident with our results. However, now that these isolates have been characterized, screening for genetic mutations is an obvious next step, although we believe that this is beyond the scope of this investigation. 

Point 3

• The data relating OD to CFU/ml is now included as "S1 Table". 

Point 4

• Thank you for this suggestion, this has been clarified in the manuscript.

Point 5

• This is work which was previously done in our lab, this data is now provided as supplementary materials as suggested in point 3.

Point 6

• When working with Brachyspira it is widely reported that broth cultures are incubated at a lower temperature than agar cultures, I do not have an explanation for why this is required/is common practice. I would speculate that incubating plates at 42C serves to make primary cultures more selective (inhibiting organisms which don't grow at 42C), once a pure culture is obtained then lower temperatures (39C or some in the field also use 37C) can be used for broth cultures. 

Point 7

• The sentence has been clarified, we did inspect broth under the microscope for the presence of motile spirochetes.

Point 8

• These dilution ranges were selected to ensure that we assessed the optical density of a wide range of concentrations. In our experience, Brachyspira species sometimes do not grow to a sufficiently high density and so we wanted to ensure that we were able to capture dilutions less than 1:2. It was critical to have multiple data points which fell within the linear portion of the curve relating OD to concentration in order to define the equation describing this relationship.

• With respect to the minimum inoculum, we anticipated that the minimum inoculum required to start a culture would be much less than would be required for detection in a spectrophotometer. Indeed, on agar we were able to observe growth with very low CFU counts.

Point 9

• The two media which were used in this study were utilized for different phases of the investigation. 

• BJ agar is a selective, antimicrobial containing media (colistin, vancomycin, spectinomycin, spiramycin and rifampin) which is used for primary isolation from clinical samples. We also use this media routinely when working with cultures of Brachyspira. A description of 

• TSA agar was used as a base for susceptibility testing, it was important to use a media which did not contain antibiotics for this application. 

• A description of the antibiotics contained within BJ agar has been added as per the recommendation of Reviewer 1.

Point 10

• We have added a description of the way the phylogenetic tree was constructed where you suggested. 

Point 11

• For this phylogenetic tree we would like to suggest that we keep the data for the same isolates which were initially presented. The ATCC strains of B. hyodysenteriae, B. pilosicoli, B. innocens and B. murdochii were included because these are the type strains of each species; because the purpose of this comparison was to help identify the species of our study isolates we only wanted to compare to the reference strains for each. With respect to B. hampsonii 30599 and B. hampsonii 30446 - these reflect the two B. hampsonii genomevars, the figure legend has been updated to clarify the identity of these isolates.

Point 12

• Clarifying language has been added to the end of the section describing sequencing of the nox gene.

Point 13

• In susceptibility testing it is critical to ensure that a consistent, standardized inoculum is tested. Adjusting all broths to a the density reported was therefore critical to the development of the standardized method we report.

Point 14

• Thank you for this comment. The decision to use a single equation was a matter of practicality; in developing a test that we hope can be applied in a diagnostic laboratory setting it would be infeasible to have separate criteria for each species. While this does introduce a level of variability into the protocol, this is accounted for by the target range of CFU/ml (1-2 × 108 CFU/ml). This range of acceptable concentrations is recommended by the Clinical and Laboratory Standards Institute and allows for a single density of organisms (MacFarland 0.5) to be used in susceptibility testing across diverse clinical isolates (Gram-positives, negatives and anaerobes).

Points 15 and 16

• The differences in maximum OD reported for each species reflect the linear portion of the standard curve over which the equation to relate OD:CFU/ml was derived. At an OD >1 there was not a linear relationship between OD/CFU/ml. 

Point 17

• In this investigation we did not study the growth kinetics of Brachyspira, all of the measurements of culture density which were recorded were endpoints and collected for the purpose of calculating CFU/ml.

Point 18

• Prior to performing any work with broth cultures they were assessed microscopically. While we did not specifically look for clumping/aggregation, this not a phenomenon that we have previously recognized. When cultures were examined prior to measuring OD they were always fresh cultures which were motile.

Point 19

• No, as per our response to point 2 we did not assess this culture collection for the presence of resistance genes or resistance conferring mutations. While this is an obvious next step, it was beyond the scope of this study.

Point 20

• Thank you very much for raising this point. This was an oversight on our part, the sequences have been submitted to GenBank and BankIt accession numbers incorporated into the text.

---

## [Editor Report · Decision Letter 1]

19 May 2023

Antimicrobial susceptibility of Western Canadian Brachyspira isolates: Development and standardization of an agar dilution susceptibility test method

PONE-D-22-35189R1

Dear Dr. Rubin,

We’re pleased to inform you that your manuscript has been judged scientifically suitable for publication and will be formally accepted for publication once it meets all outstanding technical requirements.

Kind regards,

Marwa Ibrahim Abd El-Hamid

Academic Editor

PLOS ONE
---

## [Editor Report · Acceptance letter]

22 Jun 2023

PONE-D-22-35189R1 

Antimicrobial susceptibility of Western Canadian *Brachyspira* isolates: Development and standardization of an agar dilution susceptibility test method. 

Dear Dr. Rubin:

I'm pleased to inform you that your manuscript has been deemed suitable for publication in PLOS ONE. Congratulations! Your manuscript is now with our production department. 

Kind regards, 

on behalf of

Dr. Marwa Ibrahim Abd El-Hamid 

Academic Editor

PLOS ONE